# Tannins: A Promising Antidote to Mitigate the Harmful Effects of Aflatoxin B_1_ to Animals

**DOI:** 10.3390/toxins18010015

**Published:** 2025-12-25

**Authors:** Wenhao Sun, Ruiqi Dong, Guoxia Wang, Bing Chen, Zhi Weng Josiah Poon, Jiun-Yan Loh, Xifeng Zhu, Junming Cao, Kai Peng

**Affiliations:** 1Institute of Animal Science, Guangdong Academy of Agricultural Sciences, Aquatic Collaborative Innovation Center of Guangdong Academy of Agricultural Sciences, Guangdong Key Laboratory of Animal Breeding and Nutrition, Key Laboratory of Animal Nutrition and Feed Science in South China, Ministry of Agriculture and Rural Affairs, Guangzhou 510640, China; swh2257154312@163.com (W.S.); dd2348919@163.com (R.D.); wangguoxia@gdaas.cn (G.W.); chenbing@gdaas.cn (B.C.); zhuxifeng@gdaas.cn (X.Z.); 2College of Fisheries, Guangdong Ocean University, Zhanjiang 524088, China; 3College of Fisheries and Life, Shanghai Ocean University, Shanghai 201306, China; 4Tropical Futures Institute, James Cook University Singapore, Singapore 387380, Singapore; zhiwengjosiah.poon@my.jcu.edu.au (Z.W.J.P.); james.loh@jcu.edu.au (J.-Y.L.); 5Guangzhou Fishtech Biotechnology Co., Ltd., Guangzhou 510640, China; 6Guangdong Academy of Agricultural Sciences, Guangzhou 510640, China

**Keywords:** AFB_1_, feed, tannins, detoxication, animal health

## Abstract

Aflatoxin B_1_ (AFB_1_), a major metabolite of aflatoxin, is a highly toxic carcinogen. It frequently contaminates feed due to improper storage of feed ingredients such as corn and peanut meal, with the contamination risk further escalating alongside the increasing incorporation of plant-based proteins in feed formulations. Upon entering an organism, AFB_1_ is metabolized into highly reactive derivatives, which trigger an oxidative stress-inflammation vicious cycle by binding to biological macromolecules, damaging cellular structures, activating apoptotic and inflammatory pathways, and inhibiting antioxidant systems. This cascade leads to stunted growth, impaired immunity, and multisystem dysfunction in animals. Long-term accumulation can also compromise reproductive function, induce carcinogenesis, and pose risks to human health through residues in the food chain. Tannins are natural polyphenolic compounds widely distributed in plants which exhibit significant antioxidant and anti-inflammatory activities and can effectively mitigate the toxicity of AFB_1_. They can repair intestinal damage by increasing the activity of antioxidant enzymes and up-regulating the gene expression of intestinal tight junction proteins, regulate the balance of intestinal flora, and improve intestinal structure. Meanwhile, tannins can activate antioxidant signaling pathways, up-regulate the gene expression of antioxidant enzymes to enhance antioxidant capacity, exert anti-inflammatory effects by regulating inflammation-related signaling pathways, further reduce DNA damage, and decrease cell apoptosis and pyroptosis through such means as down-regulating the expression of pro-apoptotic genes. This review summarizes the main harm of AFB_1_ to animals and the mitigating mechanisms of tannins, aiming to provide references for the resource development of tannins and healthy animal farming.

## 1. Introduction

Aflatoxin is a secondary metabolite produced by *Aspergillus flavus*, which can grow and reproduce on grains under hygrothermal environments, thus polluting agricultural products [1,2], posing a risk to food and feed safety. So far, approximately 20 types of aflatoxins have been identified, among which AFB_1_ is considered to be the most toxic [3]. It is a class I carcinogen identified by the World Health Organization and the annual losses caused by AFB_1_ amount to billions of yuan [3,4]. The pollution of AFB_1_ in feed mainly derived from corn, peanut meal, soybean, rapeseed meal, etc. and these feedstuffs are highly susceptible to mold contamination under improper storage conditions (Figure 1). Due to the growing scarcity and high cost of fish meal, there is an increasing trend towards using plant-based proteins as feed ingredients. However, this shift also increases the risk of feed contamination by AFB_1_. According to random sampling, the contamination of feed ingredients and compound feed by AFB_1_ in China is a growing concern [5,6,7]. AFB_1_ can cause genetic mutations and chromosomal abnormalities in human and animal cells [8]. In addition, AFB_1_ increases the risk of human and animal visceral injury and suppresses immune function [9]. Currently, the main methods to alleviate the toxicity of AFB_1_ include adding functional substances and nutritional regulation. For example, smectite powder can be used as an adsorbent for mycotoxins to alleviate the impact of AFB_1_ on broilers [10], yeast cell wall can alleviate the damage of AFB_1_ to intestinal epithelial cells of broilers [11], vitamin C and turmeric powder can reduce oxidative damage by reducing oxidative stress caused by AFB_1_ [12,13], and probiotics alleviate the toxicity of AFB_1_ by regulating the intestinal flora [14,15]. In addition, regulating glutathione synthesis can reduce the oxidative damage caused by AFB_1_ [16,17]. Selenium [18], boron [19] and other trace elements can enhance the antioxidant and immune capacity of the body to improve the tolerance of animals to AFB_1_.

Tannins are natural polyphenolic compounds widely existing in the plant kingdom. Generally, common sources of tannins in animal feed include grain, forage legumes or plant-derived feedstuffs. Tannins are mainly classified into condensed tannin (also known as procyanidine), hydrolyzed tannin and phlorotannins [20,21]. Tannins have traditionally been known as “anti-nutritional factor” for monogastric animals with negative effects on feed intake and nutrient digestibility by binding to dietary proteins and digestive enzymes. Our previous studies documented that a high dose (2 g/kg) of condensed tannin and hydrolyzed tannin reduced growth and induced intestinal injury in the Chinese sea bass (*Lateolabrax maculatus*) [22,23]. However, recent research showed that a low dose of tannins improved intestinal microbial ecosystems, enhanced gut health and hence increased the productive performance of animals [20,24,25,26], owing to their beneficial biological effects.

Tannins exhibit notable biological activities, including strong antioxidant and anti-inflammatory properties both in vitro and in vivo studies [27]. Recent studies have shown that dietary supplementation with 1 g/kg of condensed tannin can reduce the deposition of AFB_1_ in sea bass (*Lateolabrax maculatus*), up-regulate the gene expression of tight junction protein gene *ZO-1* (zonula occludens-1), *Claudin-3* and *occludin*, and repair the damage of intestinal barrier function induced by AFB_1_. Additionally, it increases the relative abundance of beneficial bacteria such as *Aeromonas* and *Klebsiella* in the intestine, inhibits the proliferation of harmful bacteria, and improves overall intestinal health [28,29]. Condensed tannin can also inhibit the expression of down-regulated apoptosis genes *Bax* and *caspase-3*, tumor suppressor gene *p53*, and alleviate hepatocyte mitochondrial apoptosis [30]. Hydrolyzed tannin can down-regulate the nuclear NF-κB (nuclear factor kappa B) signaling pathway in sheep, thereby alleviating inflammation [31]. Thus, tannins may alleviate the toxicity of AFB_1_ by alleviating intestinal injury, regulating intestinal flora, increasing immune ability and inhibiting apoptosis. In this review, the harmful effects of AFB_1_ to animals and the use of tannins as a mitigation strategy are summarized. The findings provide a reference for the development of tannin-based resources and the promotion of animal health in breeding programmes.

## 2. Toxicity of AFB_1_ to Animals

Upon entering the animal organs, AFB_1_ first impairs intestinal health, induces intestinal microbiota dysbiosis, and triggers intestinal inflammation. This sequence of events leads to intestinal cell damage and increased intestinal permeability, thereby enabling AFB_1_ and LPS (lipopolysaccharide) to translocate to the liver, where they induce inflammatory responses. Ultimately, such processes result in damage to other cells throughout the animal body (Figure 2). The negative effects of AFB_1_ to animals are shown in Table 1.

### 2.1. Growth Inhibition

Growth performance is a key and aggregative evaluation indicator for the growth and health of animals. In economically important species, growth performance is directly linked to the profitability of production systems. In aquatic animals, it was reported that the addition of 2230 μg/kg AFB_1_ in their feed did not affect the growth performance of hybrid grouper (*Epinephelus fuscoguttatus* ♀ × *Epinephelus lanceolatus* ♂) [43]. However, AFB_1_ induces oxidative stress and inflammation, ultimately hindering the growth of the affected individuals. In another study, dietary inclusion of 1.0 mg/kg AFB_1_ did not affect the survival rate of *Lateolabrax maculatus*, but reduced feed intake, weight gain rate and specific growth rate, resulting in a decline in growth performance [28]. In livestock and poultry, the addition of 45 μg/kg AFB_1_ in Chinese yellow chicken feed reduced the weight gain rate and feed utilization rate of broilers aged 1 to 63 days [46], which was attributed to the reduction of energy and protein metabolism efficiency by AFB_1_, resulting in the decline of growth performance [47,48]. The addition of 2550 μg/kg AFB_1_ in feed reduces the digestibility of sheep, affecting the digestion and absorption of nutrients, thus inhibiting growth [49].

The reasons for AFB_1_ inhibiting animal growth may be attributed to several mechanisms. First, AFB_1_ reduces digestive function. For instance, Pu et al. [35] reported that when the concentration of AFB_1_ in feed exceeded 280 μg/kg, nutrient digestibility in the digestive tract of pigs decreased significantly. In addition, AFB_1_ reduces the activities of key digestive enzymes such as ether extract lipase, trypsin and collagenase, thereby impairing nutrient digestion and absorption in the intestine [41,50]. Second, AFB_1_ inhibits protein synthesis, a critical process in growth. In grouper, AFB_1_ induces protein metabolism disorder by inhibiting the expression of binding protein 1, fatty acid synthase and mTOR (mammalian target of rapamycin), ultimately leading to suppressed growth performance [28,49]. Third, AFB_1_ causes tissue and organ damage. It was documented that AFB_1_ exposure led to induced intestinal and gill tissue damage in goldfish (*Carassius auratus*), increased the metabolic demands for tissue repair and detoxification [51], thus resulting in reduced growth performance due to lack of energy necessary for growth.

### 2.2. Liver Injury

Liver is the central organ for detoxification in animals. However, it is particularly vulnerable to damage from AFB_1_ exposure, which can induce liver inflammation, apoptosis and oxidation resistance, thus compromising liver health. It is reported that AFB_1_ can induce congestion and fat deposition in the liver of laying hens, and the proportion of inflammatory cells, apoptotic cells and lipid droplets in the liver are significantly increased [52]. In rats, dietary AFB_1_ can lead to an increase of glutathione content, glutathione peroxidase activity and superoxide dismutase activity in liver, increase the expression of proinflammatory factors, and lead to inflammatory cell infiltration and oxidative stress [52]. After infection with AFB_1_, the gene expression levels of apoptotic factors *caspase-1* (cysteine protease caspase-1), *IL-1β* (inflammatory cytokines interleukin-1β) and *IL-18* (interleukin-18) in the liver of mice were significantly increased, which induced liver inflammation [38]. AFB_1_ can cause necrosis and vacuolation of hepatocytes, vacuolation of mitochondria and swelling of endoplasmic reticulum in snakehead (*Channa Argus*), thus damaging the liver health [44]. Furthermore, AFB_1_ can reduce the activities of antioxidant enzymes such as CAT (catalase) and SOD (superoxide dismutase) in the liver of *Lateolabrax maculatus*, increase the content of MDA (malondialdehyde), and cause liver lipid peroxidation damage [28].

The main causes of AFB_1_-induced liver injury may include aggravation of oxidative stress and the induction of hepatocyte death. First, oxidative stress in the liver is a key driver for various liver diseases [53]. The liver is the main site for the epoxidation of AFB_1_ to AFB_1_-8,9-epoxide, which induces gene mutation by binding with DNA, leading to liver inflammation and cancer [54]. Liu et al. [55] reported that AFB_1_ can increase the content of ROS (reactive oxygen species), promote the accumulation of bile acids in the liver, aggravate oxidative stress and inflammation, and lead to liver injury by inhibiting the expression of FXR (Farnesoid X Receptor)/fibroblast growth factor 15 signaling pathway in the intestine. Second, AFB_1_ induces hepatocyte cell death. Pyrolysis of liver cells is considered an important factor in causing liver inflammation, which can activate acute and chronic hepatitis, fibrosis and non-alcoholic hepatitis [56,57]. AFB_1_ promotes the activity of hepatocyte cyclooxygenase, activates inflammatory body NLRP3 (nod-like receptor thermoprotein domain related protein 3), induces and activates apoptosis factor caspase-1, promotes the gene expression of proinflammatory cytokines *IL-18* and *IL-1β*, and leads to cell membrane damage and cell death [58]. In addition, hepatocyte pyrosis may be induced by AFB_1_ through intestinal microbiota. The microbiota affected by AFB_1_ increases intestinal permeability by destroying the mucosal layer and tight junction proteins, leading to the translocation of LPS to the liver. Finally, the focal death signal of hepatocytes is activated [38].

### 2.3. Intestinal Injury

The intestine is the primary site for the digestion and absorption of nutrients. However, AFB_1_ exposure can severely compromise intestinal health by damaging the morphology of intestinal villi, reducing the number of epithelial cells and goblet cells. Peng et al. [29] reported that dietary supplementation of AFB_1_ led to irregular arrangement of intestinal villi of *Litopenaeus vannamei*, which was manifested by deformation of villi and reduction of villi height. The height, width and area of intestinal villi in broilers infected with AFB_1_ decreased, and the depth of crypt increased significantly [33]. Similarly in broilers, AFB_1_ induced the shedding of epithelial cells at the top of intestinal villi, partial loss of the junction complex and terminal network, and a significant reduction in the number of mitochondria and goblet cells [34]. In mice, AFB_1_ caused damage to both Goblet cells and epithelial cells in the intestinal tract fed AFB_1_, which was associated with inflammation [39]. In rats, AFB_1_ aggravates oxidative stress, causes intestinal inflammation and duodenal injury [59]. In addition, AFB_1_ can also change the composition of intestinal microbiota, induce liver injury and endanger intestinal health through the intestinal microbiota bile acid FXR axis [55].

The mechanism by which AFB_1_ induces intestinal injury may involve several interrelated pathways. First, AFB_1_ triggers intestinal oxidative stress and inflammation. AFB_1_ increases the contents of ROS and MDA in the intestine of rabbits, inhibiting the activity of antioxidant enzymes, and inducing intestinal oxidative stress [49]. In mice, AFB_1_ causes the decrease of bile salt hydrolase activity, causing the accumulation of bile acids in the intestine, thereby inducing intestinal oxidative stress and inflammation [55]. Second, AFB_1_ impairs intestinal barrier function. In AFB_1_-exposed mice, alterations to the structure of intestinal flora, along with damage to tight junction proteins and intestinal mucosal layers increases intestinal permeability and causes intestinal injury [38]. AFB_1_ has also been shown to damage intestinal cell membrane, increase intestinal permeability, and induce intestinal barrier damage through clathrin-mediated endocytosis and tight junction protein transport to the cytoplasm [60]. In broilers, AFB_1_ causes the disappearance of the junction complex in the small intestine, reduction of the number of small intestinal goblet cells, and reduction of Toll-like receptor gene expression, resulting in the impairment of small intestinal barrier function [34]. Third, AFB_1_ can induce apoptosis of intestinal cells. It has been shown that AFB_1_ induces apoptosis of intestinal cells, leading to structural damage of intestinal mucosa [4]. This effect is mediated by the increased intestinal ROS content, which contributes to oxidative stress and cellular injury [49]. Zhang et al. [40] further reported that AFB_1_ can induce intestinal mucosal structural damage by down-regulating the expression of intestinal mucosa-associated connexin genes and promoting cell apoptosis.

### 2.4. Damage Reproductive Performance

For male animals, AFB_1_ exposure has been shown to reduce the size, volume and sperm motility in bird [61]. AFB_1_ damages the mitochondrial structure of testicular germ cells and stromal cells, tissue lesions, abnormal sperm development, and is accompanied by reduced mitochondrial complex enzyme activity and oxidative stress [62]. In addition, AFB_1_ increases the proportion of abnormal sperm cell morphology, reduces testicular weight, and lowers testosterone concentration [63]. The serum testosterone, prolactin and luteinizing hormone levels of male birds were significantly decreased due to AFB_1_ induction, and testicular volume was reduced, even showing necrosis of testicular parenchyma [61]. For female animals, AFB_1_ negatively affects ovarian function and oocyte development. Exposure to AFB_1_ reduced both the volume and number of oocytes [32,64], as well as the number of resting follicles and developing follicles [65]. Additionally, AFB_1_ decreased the elimination rate of the first polar body of oocytes and interfered with the oocyte maturation cycle [66,67].

The adverse effects of AFB_1_ on reproductive performance may be primarily attributed to its ability to induce cell apoptosis. AFB_1_ causes DNA strand breaks, DNA cross-linking or DNA and protein cross-linking, which increases the sperm deformity rate [68]. AFB_1_ also enhances autophagy by aggravating testicular oxidative stress, causing down-regulation of the autophagy signaling pathway PI3K (phosphatidylinositol 3-kinase)/Akt (protein kinase B)/mTOR, ultimately damaging the testis of mice [36]. In addition, AFB_1_ can also reduce the gene expression of blood–testis barrier-related connexin by regulating the p38 mitogen-activated protein kinase pathway mediated by oxidative stress, aggravate the apoptosis of mouse testicular Sertoli cells, trigger testicular mitochondria dependent apoptosis, and lead to the damage of the blood–testis barrier [37]. Beyond inducing apoptosis, AFB_1_ also impairs organelle function. AFB_1_ causes the decrease of mitochondrial complex I–IV activity, resulting in ROS accumulation and mitochondrial damage, leading to the death of testicular cells [62]. For female animals, AFB_1_ disrupts the normal distribution of mitochondria in oocytes, resulting in insufficient energy supply and exacerbating oxidative stress, and oocyte damage [69].

### 2.5. Weaken Immunity

The immune system is composed of immune organs, immune cells and immune active substances, which has the function of recognizing and eliminating foreign objects, maintaining the stability of the internal environment and physiological balance [70]. In aquatic animals, AFB_1_ reduces the activity of immunoregulatory proteins, acyl carrier proteins and immunoglobulin M in the skin, spleen, head and kidney of grass carp (*Ctenopharyngodon idella*) [45]. Ottinger et al. [71] reported that the proliferation of rainbow trout (*Oncorhynchus mykiss*) lymphocytes was inhibited by AFB_1_ and immunoglobulin decreased. In poultry, AFB_1_ exposure increases the expression of pro-inflammatory factor interferon γ in the jejunum of broilers and induces a decrease in the expression of anti-inflammatory factor interleukin 10 in the liver, resulting in intestinal and liver damage in broilers [72]. It has been reported that AFB_1_ can cause inflammatory cell infiltration in the livers of rats [59], increase the number of inflammatory cells in the livers of chickens and induce intestinal inflammation in rabbits [42,52], thus causing varying degrees of damage to the immune system.

The decrease of immune function by AFB_1_ may be related to the injury of immune organs and immune cells. The gut, liver, spleen, bursa of Fabricius, macrophages, etc. are important defense lines and components of the immune system [45,73,74]. AFB_1_ can cause cell oxidative damage, lead to liver lipid peroxidation, and ultimately suppress immune function [75,76]. When the NF-κB signaling pathway of grass carp (*Ctenopharyngodon idella*) infected with AFB_1_ was activated, the gene expression of pro-inflammatory factors in spleen and kidney were increased, the expression of anti-inflammatory factors were decreased, and the immune ability of skin, spleen and kidney were decreased, leading to damage to these organs [45]. AFB_1_ causes mitochondrial respiratory chain damage and induces oxidative stress in macrophages, which leads to inflammatory response activation and phagocyte damage [77]. Also, AFB_1_ can induce bursa tissue damage, cell cycle arrest and mitochondrial apoptosis in broilers, thereby damaging the immune system [73,76]. The decline of immune function in broilers is directly related to AFB_1_-induced DNA damage and increased thymocyte apoptosis mediated by mitochondria and the death receptor pathway [75].

### 2.6. Remaining in Animal Products

Meat, eggs and milk are all important protein sources for humans, but AFB_1_ can accumulate in the body and animal products through the food chain, causing pollution and threatening food safety [78,79,80]. In dairy cows, AFB_1_ can be detected in milk within 12 h after feeding the AFB_1_-contaminated diet. After continuous feeding for 7 d, the concentration of AFB_1_ tends to be stable, but it can still be detected in milk [79]. After feeding Nile tilapia (*Oreochromis niloticus*) with 2 μg/kg of AFB_1_ in the diet for 14 weeks, the content of AFB_1_ in muscle was 21.18 μg/kg [81]. When fed a diet containing 6400 μg/kg of AFB_1_, the content of AFB_1_ in liver and muscle of broilers was 6.97 ng/g and 3.27 ng/g, respectively [82]. Peng et al. [83] added 1 mg/kg of AFB_1_ to the diet to feed *Lateolabrax maculatus* for 56 d, and the detected concentration of AFB_1_ in muscle was 0.02 μg/kg.

The accumulation of AFB_1_ in the body may be related to its barrier penetration and metabolic transformation. AFB_1_ destroys the intestinal barrier by compromising intestinal integrity, damaging the mucosal layer or regulating inflammatory factors [82]. AFB_1_ can also destroy the blood–brain barrier and kill microvascular endothelial cells [83]. Besides, AFB_1_ can enter the brain tissue of zebrafish through the blood–brain barrier, resulting in nerve regulation injury and lipid metabolism disorder [84]. In addition, after AFB_1_ enters the body, it can be further activated by phase I drug metabolism enzymes (such as cytochrome P450 system) to form highly toxic products, which can bind with DNA or protein, exist in the organ and cause damage to the body [85].

## 3. Potential Mechanism of Tannins in Alleviating the Toxicity of AFB_1_

Tannins mitigate AFB_1_ toxicity probably by alleviating AFB_1_-induced intestinal inflammation and cellular damage, which is achieved through restoring the stability of intestinal flora, activating antioxidant and anti-inflammatory factors as well as cell signaling pathways (Figure 3). The positive effects of tannins in alleviating AFB_1_ toxicity are shown in Table 2.

### 3.1. Regulating Intestinal Health

Intestinal barrier function and intestinal flora balance are essential to maintain animal intestinal health. Tannic acid can increase the activity of glutathione peroxidase in the jejunum of piglets and up-regulate the gene expression of intestinal tight junction proteins *ZO-1*, *Claudin-3* and *occludin*, and repair intestinal damage [90]. Procyanidins can increase the relative abundance of beneficial bacteria *Bacillus* in the intestine of juvenile American eel (*Anguilla rostrata*), reduce the relative abundance of harmful bacteria *Pseudomonas* and *Aeromonas*, and maintain the balance of intestinal flora [91]. Gallnut tannic acid can reduce the relative abundance of harmful bacteria in the intestine of largemouth bass (*Micropterus salmoides*), such as *Aeromonas* and *Achromobacter*, and optimize the structure of intestinal flora [92]. The addition of 500 mg/kg tannic acid in feed increased the height of intestinal villi and the depth of crypt in broilers, and increased the activities of intestinal superoxide dismutase and glutathione peroxidase, thereby alleviating the intestinal injury caused by AFB_1_ [89]. Adding 1 g/kg of condensed tannin to the diet of *Lateolabrax maculatus* can up-regulate the gene expression of intestinal tight junction proteins (i.e., *ZO-1*, *Claudin-3* and *occludin*), and repair the intestinal barrier function damage induced by AFB_1_ [78]. Peng et al. [87] reported that the addition of 1 mg/kg of AFB_1_ to the diet increased the relative abundance of *Aeromonas* and *Klebsiella* in the intestinal tract of *Lateolabrax maculatus*, while the addition of 1 g/kg of condensed tannin could inhibit the proliferation of harmful bacteria. The addition of tannic acid in feed can increase the relative abundance of lactic acid bacteria in the intestinal tract of broilers, protect the host intestinal tract from pathogenic bacteria by synthesizing bacteriocin, and alleviate the toxicity of AFB_1_ [89,93].

### 3.2. Activating Antioxidant and Immune Signaling Pathways

Antioxidation and immunity constitute an important part of defense, which are interrelated and play a major role in maintaining body health and preventing diseases [93]. Peng et al. [29] reported that condensed tannin can activate the Nrf2 (nuclear factor erythroid 2-related factor 2) signaling pathway in the hepatopancreas of *Litopenaeus vannamei*, and up-regulate the gene expression of SOD and GPX4 (glutathione peroxidase 4) to improve the antioxidant capacity of shrimp. Similar studies have also been carried out in the spleen of *Ctenopharyngodon idella* and in the liver of *Lateolabrax japonicus.* For instance, condensed tannin up-regulated the gene expression of antioxidant enzymes by activating the Nrf2 antioxidant signaling pathway and alleviated the oxidative damage caused by AFB_1_ [94,95]. In the jejunum of lambs, condensed tannin induced a decrease in the gene expression levels of antioxidant factors glutathione peroxidase 1 and GPX4, and increased the gene expression levels of CAT and glutathione peroxidase 2 in the ileum, thereby alleviating oxidative damage in the small intestine of lamb [31]. Proanthocyanins reduce AFB_1_-induced DNA damage, decrease the frequency of micronuclei and DNA strand breaks in the bone marrow cells of rat, and restore the expression levels of tumor proteins, thus alleviating AFB_1_-induced DNA oxidative damage in rats [88]. Hydrolyzed tannins exhibit effective anti-inflammatory effects by down-regulating the gene expression levels of the NF-κB signaling pathway and mitogen-activated protein kinase in RAW 264.7 cells, thereby enhancing cellular immunity [96]. Related studies have shown that grape seed anthocyanins reduce the secretion of pro-inflammatory cytokines by inhibiting inflammation and the NF-κB signaling pathway, while activating the Nrf2 pathway and up-regulating the gene expression levels of heme oxygenase-1, quinone oxidoreductase 1, and glutamate cysteine ligase, enhancing the antioxidant capacity of broiler chickens and alleviating oxidative stress and immune damage induced by AFB_1_ [47,86].

### 3.3. Inhibiting Cell Apoptosis

Apoptosis is a fundamental biological phenomenon of cells, which is a programmed cell death phenomenon that occurs in multicellular organisms. It has been shown that supplementation of hydrolyzed tannin in diet can reduce mitosis and cell apoptosis in the cecum and colon of pigs [97]. Proanthocyanins alleviate liver cell pyroptosis by clearing ROS and inhibiting the activation of inflammasome NLRP3 in the liver [98]. Tannic acid inhibits the phosphorylation of mitogen-activated protein kinase in SH-SY5Y (human neuroblastoma cells) through the PI3K/Akt/Ntf2 signaling axis, thereby suppressing cell apoptosis [99]. Tannic acid can bind to TNF-α (tumor necrosis factor alpha) in the liver of grass carp (*Ctenopharyngodon idella*), and inhibit apoptosis of liver cells by suppressing the TNF-α signaling pathway [100]. Yulak et al. [101] reported that tannic acid inhibits H_2_O_2_-induced oxidative damage in SH-SY5Y cells, thereby alleviating cell apoptosis. AFB_1_ induces mitochondrial apoptosis in broiler liver cells by up-regulating the gene expression levels of *Bax*, *caspase-3*, and tumor suppressor gene *p53*. When anthocyanins are added to broiler feed, these gene expression levels are significantly down-regulated, indicating that anthocyanins inhibit liver cell apoptosis caused by AFB_1_ [30]. It is worth noting that these studies are based on specific animal models and specific tannin types; further study should be more precise in distinguishing between species-specific or compound-specific effects versus general phenomena when clarifying the exact mechanism of tannins.

## 4. Conclusions and Future Perspectives

AFB_1_ is a globally recognized Class I carcinogen that can transmit pollution to feedstuffs through crops or the environment, thereby endangering farmed animals. The harm of AFB_1_ to animals includes growth inhibition, liver and intestinal injury, reproductive performance and immune function reduction, and its accumulation in animal products, thereby threatening animal health and food safety. Tannins are natural polyphenolic compounds widely existing in the plant kingdom, with significant biological activities such as antioxidant and anti-inflammatory properties, and stress resistance. In recent years, it has been found that tannins can repair the damage caused by AFB_1_ in animals, demonstrating good detoxification effects. Their pathways of action include regulating animal intestinal health, activating cellular antioxidant and immune signaling pathways, and inhibiting cell apoptosis. However, the molecular mechanisms by which tannins alleviate AFB_1_ toxicity need further in-depth research. This article summarizes the harm of AFB_1_ to animals and the mitigating mechanisms of tannins, providing reference for the development of tannin resources and healthful farming.

Although significant progress has been made in research on the mitigation of AFB_1_ toxicity by tannins, several key areas require further in-depth exploration. Specifically, the structure–activity relationships of different types of tannins, their interaction mechanisms with AFB_1_, as well as their metabolic processes and active forms in various animal species remain unclear. At the molecular level, the precise regulatory targets of pathways such as Nrf2 and NF-κB need further investigation, while studies on the synergistic effects and compatibility with probiotics, minerals, and other substances are insufficient. Additionally, it is necessary to determine the safe dosage thresholds for different animal species and further explore the impacts of feed processing, storage, and environmental factors on the stability of tannins. Comprehensive research across multiple dimensions will facilitate the standardized application of tannins in healthy animal farming.

## Figures and Tables

**Figure 1 toxins-18-00015-f001:**
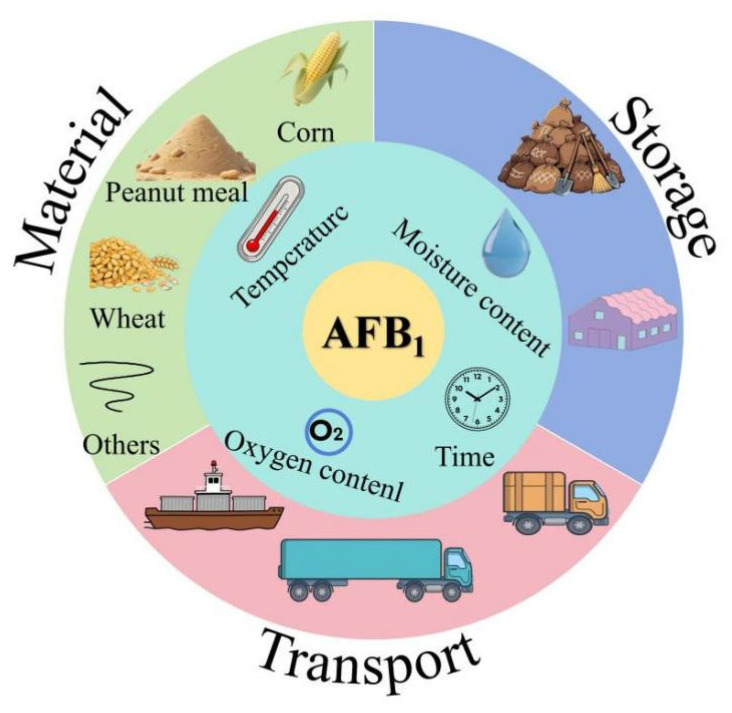
The sources and generation pathways of AFB_1_.

**Figure 2 toxins-18-00015-f002:**
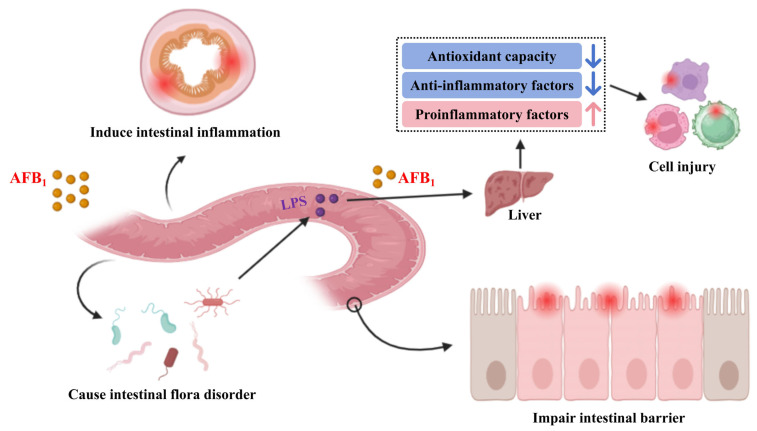
Potential toxicity of AFB_1_ to animals.

**Figure 3 toxins-18-00015-f003:**
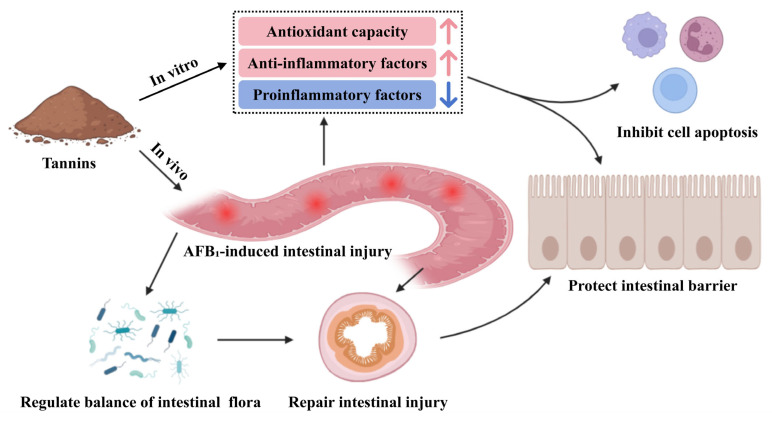
Tannins alleviate the toxicity of AFB_1_.

**Table 1 toxins-18-00015-t001:** Negative effects of AFB_1_ on animals.

Species	Body Weight/Age	Dosage of AFB_1_	Negative Effects	References
Layer Chickens	7-day-old (age)	3.91 mg/kg	Reduced growth performance, delayed sexual maturity, decreased egg-laying efficiency, hepatic inflammation accompanied by necrosis, bursa of fabricius edema	[32]
Broiler Chickens	7-day-old (age)	2 mg/kg	Reduced growth performance, decreased jejunal villus height and absorptive area, increased the relative abundance of gram-negative bacteria in the ileum	[33]
	1-day-old (age)	0.6 mg/kg	Reduced number of intestinal microvilli, caused mitochondrial vacuolization, caused the disappearance of mitochondrial cristae, junctional complexes, and terminal webs	[34]
Pig	38.21 ± 0.45 kg (body weight)	280 μg/kg	Reduced growth performance and digestibility, impaired intestinal barrier function, decreased antioxidant capacity, increased the production of pro-inflammatory factors	[35]
Mouse	21 ± 1 g (body weight)	1.5 mg/kg body weight	Oxidative stress, increased cell death and autophagy, testicular damage	[36]
	21 ± 1 g (body weight)	1.5 mg/kg	Reduced expression of connexins and promoted apoptosis of mouse testicular cells resulting in damage to the blood-testis barrier	[37]
	17.5 ± 2.5 g (body weight)	25 μg/kg body weight	Altered composition of the intestinal microbiota, which directly participates in the induction of hepatocyte pyroptosis and inflammation	[38]
	28 g (body weight)	50 μg/kg body weight	Induced colitis through interference with the AHR/TLR/STAT3 signaling axis	[39]
Sheep	50 ± 2.5 kg (body weight)	75 μg/kg	Reduced growth performance, disrupted nitrogen balance, decreased antioxidant status in the blood, impaired immune function	[40]
	31.78 ± 1.81 kg (body weight)	500 μg/kg	Reduced growth performance and digestibility, along with disturbances in choline metabolism and glycerophospholipid metabolism	[41]
Rabbit	35-day-old (age)	0.3 mg/kg	Diminished growth performance, and enhanced oxidative stress and inflammatory responses	[42]
Grouper (*Epinephelus fuscoguttatus* ♀ × *Epinephelus lanceolatus* ♂)	11.59 ± 0.03 g (body weight)	2500 μg/kg	Inhibited growth, and reduced protein metabolism and lipid metabolism	[43]
*Channa argus*	7.52 ± 0.02 g (body weight)	378.6 μg/kg	Damaged liver structure, induced inflammation, oxidative stress, and cell apoptosis, and caused AFB_1_ residues to accumulate in the liver	[44]
*Ctenopharyngodon idella*	12.96 ± 0.03 g (body weight)	150 μg/kg	Damaged the liver structure, induced inflammation, oxidative stress, and cell apoptosis, and caused AFB_1_ residues to accumulate in the liver	[45]
*Lateolabrax maculatus*	2.9 ± 0.02 g (body weight)	1.0 mg/kg	Impaired intestinal integrity, induced liver damage and intestinal flora disorder, and caused AFB_1_ residues to exist in muscle	[28]

♀ represented as a female gender and ♂ represented as a male gender.

**Table 2 toxins-18-00015-t002:** Positive effects of tannins in alleviating AFB_1_ toxicity.

Tannin Types	Species	Body Weight/Age	Dosage	Positive Effects	References
Condensed tannins	Chicken	1-day-old (age)	1 mg/kg of AFB_1_ + 250 mg/kg of proanthocyanidin	Improved growth performance, antioxidant capacity, serum biochemical parameters, immune function, and liver health, and reduced residual AFB_1_ in the liver	[47]
		1-day-old (age)	1 mg/kg of AFB_1_ + 250 mg/kg of proanthocyanidin	Attenuated immunotoxicity and oxidative stress via the NF-κB and Nrf2 signaling pathways	[86]
		1-day-old (age)	1 mg/kg of AFB_1_ + 250 mg/kg of proanthocyanidin	Enhanced activity of antioxidant enzymes and alleviated cell apoptosis	[30]
	*Lateolabrax maculatus*	2.9 ± 0.1 g (body weight)	1 mg/kg of AFB_1_ + 1 g/kg of condensed tannin	Improved immunity, alleviated liver damage, and reduced the residual amounts of AFB_1_ in the liver and muscle	[78]
		2.9 ± 0.1 g (body weight)	1 mg/kg of AFB_1_ + 1 g/kg of condensed tannin	Repaired intestinal villus damage, improved intestinal permeability, and reduced the relative abundance of Pseudomonas in the intestine	[87]
	Mouse	13 weeks-old (age)	1 mg/kg of AFB_1_ + 200 mg/kg of proanthocyanidins	Exhibited antioxidation, promoted DNA repair, and increased DNA expression	[88]
Tannic acid	Chicken	1-day-old (age)	120 μg/kg of AFB_1_ + 500 mg/kg of tannic acid	Improved growth performance, enhanced antioxidant capacity, and repaired intestinal damage	[89]

## Data Availability

No research data is used in this article, and all sources of information are linked with Digital Object Identifiers (DOIs) and URLs in the reference list.

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
