# Peer review of "Tannins: A Promising Antidote to Mitigate the Harmful Effects of Aflatoxin B_1_ to Animals"

_toxins, 2025, doi:10.3390/toxins18010015_

Round 1
Reviewer 1 Report
Comments and Suggestions for Authors
The manuscript addresses a very interesting and relevant topic: aflatoxin and approaches for reducing its harmful effects on animal organisms. The references used are current and appropriately applied.
Specific comments:
-
Latin names of fish:
Throughout the text, it would be helpful if the authors clarified which Latin names correspond to fish species by mentioning the common name before the scientific name. -
Line 70:
The sentence would be clearer if it explicitly stated that gene regulation and expression occur within the organisms. -
Tables – “Weight/Age” column:
In both tables, the “Weight/Age” column is confusing—for example, listing a pig with a weight of 38.21 ± 0.45 g. The authors should clarify whether these represent weight, age, or another parameter, and ensure realistic units. -
Tannin section – introduction:
At the beginning of the tannin section, it would be useful to include a brief description of the common sources of tannins in animal feed. -
Figure 2:
It is unclear where and how tannins influence antioxidant and anti-inflammatory factors. The figure should be revised or expanded to clearly show these relationships.
Author Response
Dear editor/reviewers,
Uploaded are our revised manuscript (ID: toxins-4035559) and our point-by-point responses to the reviewers’ comments. We thank the reviewers for their constructive comments which definitely improved the quality of the manuscript. We have carefully considered the reviewers’ comments during the revision. We also made revisions to where we identified as errors and we thought that would improve the clarity and readability throughout the manuscript. All these revisions are made in red font in the submitted copy. We hope the revised manuscript is acceptable for publication by the journal.
Please contact me if you have any further questions.
Best regards,
Correspondence: Kai Peng (pengkai@gdaas.cn)
Comments and Suggestions for Authors (reviewer 1)
The manuscript addresses a very interesting and relevant topic: aflatoxin and approaches for reducing its harmful effects on animal organisms. The references used are current and appropriately applied.
Response (R): we thank the reviewer for this positive comment.
Specific comments:
Latin names of fish:
Throughout the text, it would be helpful if the authors clarified which Latin names correspond to fish species by mentioning the common name before the scientific name.
R: Latin names of fish throughout the text were added.
Line 70:
The sentence would be clearer if it explicitly stated that gene regulation and expression occur within the organisms.
R: sentence has been revised.
Tables – “Weight/Age” column:
In both tables, the “Weight/Age” column is confusing—for example, listing a pig with a weight of 38.21 ± 0.45 g. The authors should clarify whether these represent weight, age, or another parameter, and ensure realistic units.
R: changes have been made to address this comment.
Tannin section – introduction:
At the beginning of the tannin section, it would be useful to include a brief description of the common sources of tannins in animal feed.
R: information has been added.
Figure 3:
It is unclear where and how tannins influence antioxidant and anti-inflammatory factors. The figure should be revised or expanded to clearly show these relationships.
R: figures have been revised to reflect this comment.
Reviewer 2 Report
Comments and Suggestions for Authors
Dear Authors,
Firstly, I would like to congratulate the authors on the thematic conception!
General Comments: This is a highly interesting proposal and a promising direction for the review. The material is well-organized, and the authors have done a commendable job of selecting a relevant and timely topic.
However, while the manuscript has strong potential, there are specific areas regarding the interpretation of the data and the balance of the discussion that require revision to ensure scientific rigor.
Specific Concerns:
This review frequently cites studies on specific animal models (e.g., grass carp, broilers, and rats) and specific tannin types (e.g., proanthocyanidins and tannic acid) to support broad claims about tannins and animals in general. For instance, the discussion on how proanthocyanidins downregulate Bax and caspase-3 in broilers is a valid specific finding, but assuming that this exact mechanism applies universally to all tannin types and animal species is a potential oversimplification. The manuscript should be more precise in distinguishing between species-specific or compound-specific effects versus general phenomena.
Reliance on in vitro data: A significant portion of the mechanistic evidence provided relies on in vitro studies (e.g., using RAW 264.7 or SH-SY5Y cells). While these are essential for mapping molecular pathways, results from cell cultures do not always translate directly to whole living organisms with complex digestive and metabolic systems. The leap from cell-line effects to systemic benefits in farmed animals is significant and is not fully bridged in the current study.
Balance of anti-nutritional effects: The primary focus of the review leans heavily towards the positive, detoxifying effects of tannins. While the potential for high tannin concentrations to act as anti-nutritional factors (ANFs)—specifically by binding to dietary proteins and digestive enzymes—is briefly mentioned in the context of dosage, it has not been sufficiently explored. To provide a complete risk-benefit analysis, a more balanced discussion is needed that weighs these well-known negative effects against the proposed benefits.
Question for the Authors
Considering the known bioavailability issues of tannins and their tendency to form complexes with proteins in the gastrointestinal tract, how do you reconcile the strong efficacy observed in the in vitro cell models with the potential reduction in efficacy and anti-nutritional risks in in vivo livestock feeding trials?
Author Response
Dear editor/reviewers,
Uploaded are our revised manuscript (ID: toxins-4035559) and our point-by-point responses to the reviewers’ comments. We thank the reviewers for their constructive comments which definitely improved the quality of the manuscript. We have carefully considered the reviewers’ comments during the revision. We also made revisions to where we identified as errors and we thought that would improve the clarity and readability throughout the manuscript. All these revisions are made in red font in the submitted copy. We hope the revised manuscript is acceptable for publication by the journal.
Please contact me if you have any further questions.
Best regards,
Correspondence: Kai Peng (pengkai@gdaas.cn)
Comments and Suggestions for Authors (reviewer 2)
Dear Authors,
Firstly, I would like to congratulate the authors on the thematic conception!
General Comments: This is a highly interesting proposal and a promising direction for the review. The material is well-organized, and the authors have done a commendable job of selecting a relevant and timely topic.
However, while the manuscript has strong potential, there are specific areas regarding the interpretation of the data and the balance of the discussion that require revision to ensure scientific rigor.
R: we thank the reviewer for this positive comment and these constructive comments.
Specific Concerns:
This review frequently cites studies on specific animal models (e.g., grass carp, broilers, and rats) and specific tannin types (e.g., proanthocyanidins and tannic acid) to support broad claims about tannins and animals in general. For instance, the discussion on how proanthocyanidins downregulate Bax and caspase-3 in broilers is a valid specific finding, but assuming that this exact mechanism applies universally to all tannin types and animal species is a potential oversimplification. The manuscript should be more precise in distinguishing between species-specific or compound-specific effects versus general phenomena.
R: the reviewer raises a good question. We completely agree the reviewer’s comment that it exists the species-specific or compound-specific effects as reflected by different animal models and tannin types. Tannins represent a class of compounds, so far it is hard to clarify the exact mechanism or make clear that how tannins exert their biological effects by a specific finding. In fact, even for the same type of tannins, the mechanism of its effect on different animals may also be different. We thank the reviewer’s kind remind/suggestion on distinguishing between species-specific or compound-specific effects versus general phenomena. We emphasize this viewpoint in the text.
Reliance on in vitro data: A significant portion of the mechanistic evidence provided relies on in vitro studies (e.g., using RAW 264.7 or SH-SY5Y cells). While these are essential for mapping molecular pathways, results from cell cultures do not always translate directly to whole living organisms with complex digestive and metabolic systems. The leap from cell-line effects to systemic benefits in farmed animals is significant and is not fully bridged in the current study.
R: we thank the reviewer for this comment. The reviewer pointed out the problems in the current research, and this is precisely what we were concerned about. Most of mechanistic evidence relies on in vitro studies but is was not always been verified in vivo. This explains why the in vivo studies we have found is limited.
Balance of anti-nutritional effects: The primary focus of the review leans heavily towards the positive, detoxifying effects of tannins. While the potential for high tannin concentrations to act as anti-nutritional factors (ANFs)—specifically by binding to dietary proteins and digestive enzymes—is briefly mentioned in the context of dosage, it has not been sufficiently explored. To provide a complete risk-benefit analysis, a more balanced discussion is needed that weighs these well-known negative effects against the proposed benefits.
R: we thank the reviewer for this comment. We agree the reviewer’s comment that tannins have their another mask called anti-nutritional factors. We did not shy away from mentioning in the text that the anti-nutritional effects of high doses of tannins. This review aims to emphasize the beneficial effects of low-dose tannins, which are often overlooked by us. In fact, our previous studies have reported tannins’ anti-nutritional or negative effects on fish (e.g., Qiu et al., 2024. Dietary condensed tannin exhibits stronger growth-inhibiting effect on Chinese sea bass than hydrolysable tannin. Anim. Feed Sci. Technol. 308, 115880. Chen et al., 2022. Condensed tannins increased intestinal permeability of Chinese seabass (Lateolabrax maculatus) based on microbiome-metabolomics analysis. Aquaculture 560, 738615.). To ensure the smoothness of the logical structure of this review, we did not mention too many negative effects of tannins. We have added some discussion about tannins’ anti-nutritional effects to address this comment.
Question for the Authors
Considering the known bioavailability issues of tannins and their tendency to form complexes with proteins in the gastrointestinal tract, how do you reconcile the strong efficacy observed in the in vitro cell models with the potential reduction in efficacy and anti-nutritional risks in in vivo livestock feeding trials?
R: the reviewer raise a good question. Tannin is a “double-edged sword” with high-dose of tannin has the potential reduction in efficacy and anti-nutritional risks, whereas low-dose of tannin shows beneficial effects. Most of our previous in vitro/ in vivo studies including the reports in this review are based on low-dose of tannins. In this case, the side-effect of tannins can be disregarded.